# The Impact of Green Mergers and Acquisitions on Corporate Environmental Performance: Evidence from China's Heavy-Polluting Industries

Yingying Xu , Wen Wang *, Honggui Gao and Huaxiong Zhu

School of Economics, Zhongnan University of Economics and Law, Wuhan 430073, China; jasmine20210901@126.com (Y.X.)
* Correspondence: 15729386612@163.com; Tel.: +86-13610322045

**Abstract:** This study examined the impact of green mergers and acquisitions (green M&As) on corporate environmental performance. Applying the Differences-in-Differences (DID) model to a sample of Chinese heavy-polluting-industry companies listed on the Shanghai and Shenzhen stock exchanges from 2010 to 2022, our study results show that the adoption of green M&As by the listed Chinese heavy polluters can lower corporate environmental capital expenditure and significantly improve corporate environmental performance. Meanwhile, the positive effects of green M&As on environmental performance are also found to be stronger for state-owned enterprises, young enterprises, and enterprises located in areas with low financial investments in energy efficiency and environmental protection, according to a heterogeneity study conducted for this paper. The analysis of mediating effects shows that the green M&A of heavily polluting firms will have a catalytic effect on the improvement of firms' environmental performance by promoting their green technological innovation and, in turn, their environmental performance. Furthermore, the moderating effect analysis demonstrates that the quality of the firm's internal controls and the CEO's prior environmental experience are both factors that can support the beneficial impact of green M&A on the enhancement of the firm's environmental performance. This paper enriches the theoretical research system of green M&A and green investment driving mechanisms, and at the same time provides empirical support and strategic reference for the green strategy decision of heavy-polluting enterprises.

**Keywords:** green mergers and acquisitions; environmental performance; green innovation; heavy polluters; DID

## 1. Introduction

China's economy has grown significantly over the past 40 years of reform and opening up, but the environmental impact of its long-standing, crude economic model has increased [1,2]. Heavily polluting enterprises are a major source of environmental pollution and energy consumption, but they are also an important support for China's real economy as they are involved in industries that have a bearing on the country's economy and people's livelihoods, such as iron and steel, thermal power, petrochemicals, pharmaceuticals and textiles [3,4]. Green M&As are one of the main approaches to corporate green management. M&As are carried out by enterprises to acquire green technologies and equipment and improve energy saving and emission reduction capabilities, and achieving green transformation is defined as green M&As. The 'eye-catching effect' and speed advantage of green M&As compared to self-improvement of corporate environmental performance makes companies not supported by industrial policies highly inclined to green M&As [5]. As a result, heavy polluters are likely to focus on green M&As to improve their environmental performance quickly due to their pressing need to reduce pollution [6,7]. However, academic research on green M&As focuses only on the motives of green M&As

by enterprises to alleviate regulatory pressure and maintain their image and reputation, but fails to discuss the effects and contributions of green M&As in depth. This type of green M&A with the goal of acquiring green technologies, resources, and outcomes has received little attention in the current literature on environmental performance and M&As, and very little of the literature is concentrated on heavy polluters.

Can green M&As achieve synergistic development of enterprise emission reduction and economic growth to support the successful transformation of China's industrial structure and long-term stable economic development? This paper empirically examines the impact of green M&As on corporate environmental performance using panel data of companies in China's heavy pollution industry. We test for firm heterogeneity in three dimensions: firm equity, firm age, and government environmental spending in the region where the firm is located. It also provides an in-depth analysis of the impact mechanisms and transmission paths, examining the mediating mechanisms of corporate green technological innovation, as well as the moderating mechanisms of the quality of corporate internal controls and the environmental experience of corporate CEOs.

The main contributions of this paper are as follows: Firstly, the existing literature tends to focus on the motivations of firms to carry out green M&As and ignores the impact of green M&As. We examine the objective performance of green M&As affecting environmental performance of heavily polluting firms, expand the research on the influencing factors of corporate green M&As and green investments, enrich the research on the driving mechanisms of corporate environmental performance, and provide micro evidence for promoting the green development of heavily polluting firms. Secondly, we adopt the DID method to reveal the relationship between green M&As and environmental performance, effectively controlling the potential endogeneity problem, circumventing the limitations of existing studies that directly adopt dummy variables to measure green M&As and conduct an ordinary regression analysis, helping to improve the accuracy of the research findings. Thirdly, we provide useful advice for companies to formulate their growth strategies; meanwhile, the conclusions provide a theoretical basis and policy rationale for how the relevant government departments can further improve the governance mechanism for the corporate environment and promote the harmonious development of the environment and the economy, and also provide an effective reference for other developing countries.

## 2. Literature Review and Hypothesis Development

### 2.1. Literature Review

In order to promote the strategic deployment of building a beautiful China, how to improve the environmental performance of heavy-polluting enterprises at the micro level has become an important issue of concern to the Chinese government, society, and enterprises at present. Studies have shown that there is a distinction between broad and narrow environmental performance of enterprises, with broad environmental performance referring to the efforts and effectiveness of enterprises in pollution prevention and control, effective use of resources, and reduction of environmental risks [8], while narrow environmental performance refers to a system of indicators that can be identified and measured by a company through quantitative criteria, for example, the quantitative levels of solid, liquid, gaseous and other types of harmful substances emitted by enterprises in the course of production and operation [9]. A unified system for measuring environmental performance has not yet been established in academia, and studies have measured the environmental performance of enterprises in terms of environmental investment, pollutant emissions, emission fees, and the existence of environmental violations [10–13]. Meanwhile, in terms of factors influencing corporate environmental performance, some studies have shown that corporate environmental performance is influenced by a range of factors including media attention [14], government regulation [15,16], industry competition [17], corporate technological innovation [18,19], corporate productivity [20], corporate internal management structure and environmental management philosophy [21,22].

With the promulgation of the Environmental Protection Law of the People's Republic of China, how to improve the environmental performance of enterprises and ensure that green production and economic profits of heavy-polluting enterprises develop together has been a challenge to the success of the green transformation of Chinese industries. According to Porter and Vander Linde [23], in the long term, business development and environmental management are mutually beneficial since the cost savings of enterprises due to technological improvements will offset the increased costs due to green investments. However, in the short term, given the rapidly changing market and limited capital, increasing production costs and foregoing quality opportunities for re-expansion in order to achieve environmental goals will seriously reduce productivity and constrain business development [24–26]. In the face of growing social awareness of environmental issues, companies are looking to find and adopt a gradual transformation that balances growth and environmental protection [27,28]. Based on Williamson's [29] theoretical idea of mergers and acquisitions to acquire cutting-edge technological resources, improve their technological content and achieve technological change, scholars have clearly proposed the concept of green mergers and acquisitions to combine the urgent green transformation needs of heavy-polluting enterprises with the concept of "green" mergers and acquisitions to acquire green technology and equipment and other resources, improve energy saving and emission reduction capabilities, and achieve green transformation [30,31]. The concept of a green M&A is defined by scholars as an M&A that is carried out by enterprises for the purpose of acquiring green technologies and equipment, improving energy saving and emission reduction capabilities, and achieving green transformation [32–35]. Through the introduction of green targets, companies can not only directly improve their production pollution situation, but also accelerate the emergence of a win–win situation for both profitability and environmental protection through the catalytic effect of new technologies and talents [36]. In recent years, more and more heavily polluting enterprises have realized transformation and development through green M&A activities or sent good signals to the market to protect the environment and clean production, and green M&As with environmental protection themes have gradually become a hotspot of attention in the capital market [33,35]. At the same time, once a company has completed its internal capital accumulation, M&As, to a certain extent, can serve as an important way for the company to expand its scale and improve its resource allocation efficiency and competitiveness [5,37]. Green M&As are one of the main approaches to corporate green management and are a broad integration of corporate technology M&As and environmental protection concepts, which are still in their infancy. Initial studies have examined the drivers of green M&As and have identified the influence of environmental regulation, social opinion and internal managerial traits on green M&As [37–39].

In summary, academic research on green M&As is still in its early stages, focusing only on the motives of green M&As by enterprises to alleviate regulatory pressure and maintain their image and reputation, but failing to discuss the effects and contributions of green M&As in depth. Therefore, based on a sample of Chinese heavy-polluting enterprises, this paper expands the research on green M&As and green investment driving mechanisms, and through a series of tests, effectively identifies the effects of green M&As on the environmental performance of heavy-polluting enterprises, providing theoretical references and micro evidence for the green development of heavy-polluting enterprises and their strategic decision-making choices.

### 2.2. Hypothesis Development

Enterprises are the micro-foundation of economic operation and the main body of production and operation in the process of economic development [40]. For many years, heavy-polluting industries such as thermal power generation and iron and steel have contributed greatly to China's rapid economic development, but the negative environmental externalities arising from their production processes have also become a barrier to improving corporate environmental performance [41]. Since environmental protection has

been incorporated into the assessment and selection of China's local government leaders and cadres for promotion, environmental policy regulations have been strengthened and pressure on heavily polluting enterprises to rectify environmental problems has increased. Under strict government regulation and public scrutiny [2], heavily polluting enterprises that fail to meet emission reduction requirements are not only unable to enjoy policy dividends but may also face high penalties [40,42].

This predicament particularly challenges traditional industrial enterprises, which rely heavily on resources, emit high pollution levels and have weak innovation bases [43]. In order to improve the speed and quality of environmental management, heavy polluters often have only three options: shutting down and reducing production, increasing internal green investment, or implementing green mergers and acquisitions [44]. Shutting down and reducing production means that the company's economic efficiency will be hit hard in the short term, which will likely lead to a series of serious consequences, and this strategy will not be considered unless the company has difficulty coping with the administrative efforts of environmental protection [4]. The in-house green investment will promote the improvement of green technology, which is effective in the long run for the development and green transformation of the company, but in the short term, it is more expensive, less profitable, and includes the risk of R&D failure [45,46]. Compared to the first two options, heavily polluting companies prefer the shorter and more direct green M&A approach to acquire green technologies, energy-saving equipment, and other resources of the acquired company in order to quickly reduce pollution emissions and transition to cleaner production [14]. In addition to the advantages of short lead times and direct results, green M&As have three other advantages. Firstly, through horizontal mergers and acquisitions, companies can easily improve their management efficiency by reallocating resources, and achieving scale and synergy effects [9]. Secondly, environmental protection has become a hot topic in recent years. According to the theory of attention distribution, green M&A can convey the image of low-carbon production and ecological harmony to the market and the public, demonstrating corporate commitment, gaining "eyeballs" and effectively winning the favor of the capital market [47]. Finally, green M&A practices are conducive to regional economic growth and job creation, satisfying the economic performance needs of local officials on the basis of improving the regional environment, and are also favored by local policies [9]. Overall, under the high pressure of government environmental regulation, heavy polluters will tend to enhance their corporate environmental performance through green M&A and alleviate the environmental pressure from the government.

**Hypothesis 1.** *Green M&As by firms in heavily polluting industries will promote improved environmental performance.*

So, is a green M&A a stop-gap measure forced by the situation of China's heavy-polluting companies, or is it a spontaneous act of enterprises to protect the environment? By analyzing the findings of existing studies, we believe that this question should be analyzed on a case-by-case basis. A study by Pan et al. [14] suggested that a green M&A in response to public opinion is only a strategic tool to reduce external attention, while another study by Pan et al. [47] suggested that managers who are imbued with Confucian culture can intrinsically motivate heavy polluters to implement green M&As that are both environmentally and economically beneficial. Cao and Ma [44] suggest that green M&As significantly increase firm value in the year of acquisition.

We believe that the acquisition of technology and talent through green M&As is not a once and for all solution for heavy polluters. At the same time, in the context of the full implementation of China's carbon emissions trading market, enterprises with a technological first-mover advantage will save more carbon allowances, creating an "innovation compensation effect" [39,48]. On the one hand, the country's heavy-handed approach to tackling pollution makes the consequences of non-compliance by heavily polluting companies disastrous [43]. On the other hand, there is huge potential and scope for companies to reap

economic benefits through energy saving and emission reduction [49,50]. In comparison, after acquiring resources through green M&As to build a good foundation, heavy polluters are more willing to choose a strategy that meets environmental regulation standards [36], change their previous crude development model and take the initiative to implement a green technology leadership strategy to deepen their innovation research to enhance their core competitiveness, corporate value and industry status [51,52]. Therefore, we believe that the implementation of green M&A by heavily polluting companies can drive their environmental performance through green technology innovation.

**Hypothesis 2.** *Green M&As by firms in heavily polluting industries will promote improved environmental performance through green technology innovation.*

Internal control runs through financial management, capital supervision, sales and production, information communication, and other basic business activities, and is the key to the effectiveness of the enterprise risk governance mechanism, which has a substantial impact on the scientific nature of corporate decision-making [22]. Good quality of internal control can improve management's ability to predict, guarantee the effectiveness of M&A evaluation, and reduce M&A risks [13]. Specifically, high-quality internal control is one of the outstanding manifestations of management's ability, and management with policy sensitivity, risk prevention awareness, and due diligence tends to pay more attention to the quality of internal control and is more able to notice corporate issues highlighted in internal management reports. Such executives have a long-term vision of development and are able to make more timely responses to the firm's external environmental policies [39]. Meanwhile, high-quality internal control can bring accurate internal information, effective safeguard mechanisms, and reliable accounting and financial information to the enterprise. This will provide basic information support for management to measure the enterprise's ability and predict the implementability and riskiness of the M&A [21]. In addition, under high-quality internal control, the accounting and financial information of the enterprise has stronger credibility, which will also bring convenience to the assessment work of the enterprise's external institutional investors and improve the effectiveness of external supervision [37]. Only after careful investigation and assessment and careful finalization of the merger and acquisition, can we integrate the resources of both parties more quickly after the merger and acquisition, unify the enterprise's organizational structure, culture and strategic objectives, and truly achieve the purpose of enhancing the performance of the enterprise environment [47]. Therefore, green M&As are more likely to improve the environmental performance of enterprises with good internal control quality.

**Hypothesis 3.** *Green M&As are more likely to improve the environmental performance of enterprises with good internal control quality.*

The unique experiences of business executives, as opposed to demographic characteristics, such as age and gender, can have a particular 'imprint' on their perceptions, thought patterns, values and decision-making, which are reflected in their daily work and strategic choices [53]. As environmental protection is not a quick fix, its long-term, creative and autonomous nature can fundamentally change the environmental values of those working in environmental protection and make them concerned about issues related to the environment [47]. If the main management staff of an enterprise has been engaged in or participated in environmental protection work or study, it will form an environmental protection 'imprint' [54]. This will significantly increase the environmental awareness of executives and bring a wealth of environmental knowledge and experience to the firm. The CEO is the key decision maker in corporate management, responsible for the planning and implementation of financial, strategic, operational and other important tasks. A CEO with environmental experience will have a stronger sense of social responsibility and will be able to recognize the urgency and necessity of environmental protecting problems in a

timely manner [36]. Under increasingly stringent environmental regulations and potential pollution costs, the executive team of heavily polluting companies often hesitates to make decisions on development and transformation, while CEOs with environmental experience can avoid short-sightedness, become familiar with the national environmental policy faster, reduce the risk of transformation, and highlight the advantages in the process of sustainable and green development of the enterprise [14]. Therefore, if the CEO has environmental experience, the positive effect of green M&As on corporate environmental performance will be stronger.

**Hypothesis 4.** *If the CEO has environmental experience, the positive effect of green M&As on corporate environmental performance will be stronger.*

### 3. Research Design

*3.1. Sample Selection and Data*

Using panel data of listed companies in Shanghai and Shenzhen stock exchanges from 2010 to 2022 in China's heavy pollution industry, we investigated the effects of corporate green M&As on environmental performance. We screened the sample in the following ways: (1) Excluding M&A events with failed deals, acquisition amounts less than RMB 1 million, equity acquisition ratios less than 30% or already holding more than 30% equity ratio in the target company. (2) Excluding M&A samples with failed deals or missing data. (3) Excluding ST, PT and insolvent companies. (4) Excluding M&A samples where the business type is divestiture, asset replacement, debt restructuring, or share buyback. Finally, we retained only M&A events where the transaction type was an equity acquisition. At the same time, if the same firm conducted multiple M&As in the same year, the samples with the same M&A targets are combined, and only the sample with the largest transaction amount and the highest acquisition ratio is retained for the samples with different M&A targets. After screening, we obtained a total sample of 141 firms. Figures A3 and A4 in the Appendix A report the statistical description of the sample of firms.

Meanwhile, according to the Circular of the Ministry of Ecology and Environment of the People's Republic of China on the Issuance of the Classification and Management List of Listed Companies in Environmental Protection Verification Industry, we define the following industries as heavy-polluting industries: coal mining and washing industry, oil and gas mining, ferrous metal mining and processing industry, non-ferrous metal mining and processing industry, textile industry, leather, fur, feather and its products and footwear industry, paper and paper products industry, petroleum processing, coking and nuclear fuel processing industry, chemical materials and chemical products manufacturing industry, chemical fibre manufacturing industry, rubber and plastic products industry, non-metallic mineral products industry, ferrous metals smelting and rolling processing industry, non-ferrous metals smelting and rolling processing industry, and electricity and thermal power production and supply industry. According to the Guidelines on Industry Classification of Listed Companies revised by the China Securities Regulatory Commission in 2012, the codes for the heavy-polluting industries are B06, B07, B08, B09, C17, C19, C22, C25, C26, C28, C29, C30, C31, C32 and D44.

In addition, the internal control quality data come from the DIB internal control and risk management database; the enterprise green innovation data come from the Incopat patent database; the data at the prefecture level and city level come from the China Urban Statistical Yearbook and local statistical bureaus; the data at the provincial level come from the National Bureau of Statistics, China Statistical Yearbook, China Environmental Yearbook, and China Environment Statistical Yearbook; and the rest of the data at the enterprise level come from the China Stock Market & Accounting Research Database (CSMAR) and WIND database. See Table 1 for a description of the variables.

**Table 1.** Variable definitions.

| Symbol | Variable | Definition |
|---|---|---|
| EP | Environmental performance | The natural log of the firms' environmental capital expenditure |
| Greenma | Green M&A | If the firm has implemented green M&As, take a value of 1; otherwise, take 0 |
| Size | Enterprise size | The natural log of the total enterprise assets |
| Employees | Number of employees | Number of employees in the enterprise |
| Lev | asset-liability ratio | Total liabilities/total assets |
| Roa | ROA | Net profit/average total assets |
| Roe | ROE | Net profit/net assets |
| Nsale | net sales margin | Net profit/sales revenue |
| Sale | Sales revenue growth rate | (Amount of new sales revenue-amount of original sales revenue)/amount of original sales revenue |
| CR | current ratio | Current assets/current liabilities |
| Lerner | Lerner index | (Operating income-operating costs-selling expenses-administrative expenses)/Operating income |
| Pgdp | GDP per capita | GDP per capita in the region where the enterprise is located |
| Sur | industrial structure | Share of secondary industry in the region where the enterprise is located |
| Pop | population density | Population density in the area where the enterprise is located |

### 3.2. Variable Selection

Environmental Performance (*EP*). The methods used in the literature to assess EP include the Toxic Release Inventory (TRI), rankings published by authoritative organizations, and scoring indices. However, since China does not have a sound assessment system for this indicator, the only available measurement data are enterprise pollutant emissions, sewage charges, categorical assignments of environmental incentives or penalties, and environmental capital expenditures. Among them, pollutant emissions and sewage charges at the enterprise level are seriously missing, and the assignment of environmental incentives or penalties is difficult to standardize, so they cannot become reliable assessment data. Therefore, we refer to Patten [10] and Hu et al. [11] and use the logarithmic value of corporate environmental capital expenditure to measure environmental performance. The smaller the value, the better the environmental performance of the firm. We refer to Xie [12] and use the data after normalizing the firm's environmental capital expenditure by the total assets or the balance of shareholders' equity of the firm in the robustness test section.

Green M&A (*Greenma*). Green M&A refers to the M&A implemented by enterprises adhering to the green concept for the purpose of acquiring resources such as green technology and equipment, improving energy-saving and emission reduction capabilities, and realizing green transformation. We refer to the study of Pan et al. [14], by collecting and studying the M&A announcements of heavily polluted listed companies, based on the core information such as the background status of the main merger and the enterprises of both parties and the content of the M&A, and clarifying the M&A objectives and M&A impacts of the acquirer and the acquiree. We manually screened out the M&A events that meet the definition of green M&A and assigned them the value of 1, otherwise, it is 0.

Control variables. Referring to Chan et al. [6] and Anand et al. [17], we adopt control variables in both firm and region dimensions. The enterprise level selects control variables from five perspectives: enterprise size, financial leverage, profitability and growth, asset realization, and competitive position; specifically, enterprise size (*Size*) and number of employees (*Employees*) are used to measure enterprise size; gearing ratio (*Lev*) is used to measure enterprise financial leverage; return on assets (*Roa*), return on net assets (*Roe*), net sales margin (*Nsale*) and sales revenue growth rate (*Sale*) to measure corporate profitability and growth; current ratio (*CR*) to measure the ability of corporate assets to liquidate; and Lerner index (*Lerner*) to measure the competitive position of the enterprise. At the same time, the M&A of enterprises is also closely related to the degree of economic development of the region in which they are located, and in order to enhance inter-regional comparability,

the characteristics of regional economic heterogeneity should be further controlled. The control variables at the regional level are GDP per capita (*Pgdp*), industrial structure (*Sur*) and population density (*Pop*).

*3.3. Model Construction*

The DID model assesses the implementation effect of the policy by comparing the difference between the experimental group and the control group before and after the policy, and the DID model can effectively overcome the endogeneity problem, which is more popular in domestic and international academic circles [55]. To accurately identify the impact of green mergers and acquisitions on the environmental performance of heavy-polluting enterprises, this paper adopts the double difference method to construct model (1). If the coefficient $\alpha$ in model (1) is significantly negative, it proves that the green mergers and acquisitions of heavy-polluting enterprises can reduce the total environmental protection investment of enterprises and improve environmental performance.

$$EP_{i,t} = \alpha Greenma_{i,t} + \lambda Controls_{i,t} + \gamma_t + \mu_i + \varepsilon_{i,t} \tag{1}$$

where $i$ represents the firm and $t$ represents time. $EP_{i,t}$ represents the environmental performance. $Greenma_{i,t}$ represents whether firm $i$ has implemented green M&A at time $t$. $Controls_{i,t}$ is a series of control variables. $\gamma_t$ is the time-fixed effect, $\mu_i$ is the individual fixed effect for each firm, and $\varepsilon_{i,t}$ is the error term.

## 4. Empirical Results

*4.1. Descriptive Statistics*

Table 2 reports the descriptive statistics of the main variables in this paper. The minimum value of the dependent variable (*EP*) is 6.00, the maximum value is 21.21, and the mean value is 15.37, which is almost the same as the results of Wang and Zheng [56], who used the same type of data computation, proving the accuracy of the environmental performance data in this paper. At the same time, there are large individual differences between the data of the sample companies, which possesses the researchability of the data. The mean value of the explanatory variable (*Greenma*) is 0.08, which indicates that a total of 8% of the sample of heavily polluting firms during the sample period with an implemented green M&A. In addition, green M&A events accounted for a total of 12.3% of M&A events, indicating that green M&As gradually became one of the most important ways for heavy-polluting enterprises to achieve green development. The result is between the statistical results of the studies by Pan et al. [47], Han et al. [36], and Zhou et al. [37], which proves that the data sample is accurate and credible.

**Table 2.** Descriptive statistics.

| Variable | Mean | Std. Dev | Min | Max | Obs |
|----------|------|----------|-----|-----|-----|
| *EP* | 15.37 | 1.85 | 6.00 | 21.21 | 1833 |
| *Greenma* | 0.08 | 0.27 | 0.00 | 1.00 | 1833 |
| *Size* | 22.73 | 1.35 | 18.16 | 26.37 | 1833 |
| *Employees* | 7.67 | 10.85 | 0.01 | 108.30 | 1833 |
| *Lev* | 0.53 | 0.23 | 0.02 | 2.99 | 1833 |
| *Roa* | 0.03 | 0.09 | −1.13 | 0.74 | 1833 |
| *Roe* | −0.17 | 5.81 | −174.90 | 8.72 | 1833 |
| *Nsale* | 0.04 | 0.86 | −5.02 | 34.27 | 1833 |
| *Sale* | 0.89 | 17.99 | −0.96 | 665.50 | 1833 |
| *CR* | 1.68 | 3.05 | 0.05 | 68.97 | 1833 |
| *Lerner* | 0.08 | 0.32 | −11.71 | 0.80 | 1833 |
| *Pgdp* | 6.90 | 3.96 | 1.01 | 46.77 | 1833 |
| *Sur* | 47.18 | 10.56 | 15.60 | 74.73 | 1833 |
| *Pop* | 561.80 | 405.50 | 17.00 | 2648.00 | 1833 |

### 4.2. Benchmark Regression

Table 3 presents the results of the benchmark regressions on the impact of green M&As on the corporate environmental performance of heavily polluting listed firms in China. Column (1) of Table 3 reports the case where no control variables are added; column (2) builds on column (1) by adding firm-level and region-level control variables but not controlling for fixed effects; column (3) builds on column (2) by controlling only for time fixed effects but not individual fixed effects; column (4) builds on column (2) by controlling only for individual fixed effects but not time fixed effects; and column (5) controls not only for firm- and region-level control variables but also for time and individual fixed effects. Regardless of whether the corresponding control variables and fixed effects are controlled or only part of the control variables and fixed effects are controlled, the regression results of *Greenma* in columns (1) to (5) are significantly negative at the 1% level of significance, and the results are relatively robust, which indicates that the implementation of green M&As by the listed heavy polluting firms in China can reduce the environmental protection capital expenditures of the firms and significantly increase the environmental performance of the firms. This result is a good proof of hypothesis 1.

**Table 3.** Benchmark regression.

| Variables | (1) EP | (2) EP | (3) EP | (4) EP | (5) EP |
|---|---|---|---|---|---|
| *Greenma* | −0.645 *** | −0.954 *** | −1.053 *** | −0.894 *** | −0.968 *** |
| | (−2.628) | (−3.715) | (−4.090) | (−3.410) | (−3.736) |
| *Controls* | No | Yes | Yes | Yes | Yes |
| Individual FE | No | No | No | Yes | Yes |
| Time FE | No | No | Yes | No | Yes |
| *_cons* | 15.404 *** | 1.060 | 0.993 | 1.405 | 0.676 |
| | (124.505) | (0.571) | (0.505) | (0.515) | (0.243) |
| N | 1833 | 1833 | 1833 | 1833 | 1833 |
| $R^2$ | 0.013 | 0.121 | 0.174 | 0.128 | 0.180 |

Note: Robust t-statistics in parentheses; *** $p < 0.01$, ** $p < 0.05$, * $p < 0.1$.

We observe the control variables and find that the coefficient of gearing ratio (*Lev*) is significantly positive at the 5 percent level; the coefficients of return on net assets (*Roe*), net sales margin (*Nsale*) and sales revenue growth rate (*Sale*) are all significantly negative at the 1 percent level; and the coefficient of the Lerner index (*Lerner*) is significantly negative at the 1 percent level. It indicates that heavily polluting listed companies with lower financial leverage, better profitability and growth, and stronger corporate competitive positions have better environmental performance. The coefficients of industrial structure (*Sur*) and population density (*Pop*) are both significantly positive at the 5% level, indicating that there are regional differences in the environmental capital expenditures of enterprises and that heavily polluting listed enterprises in more industrialized and densely populated regions are more inclined to increase their environmental capital expenditures.

### 4.3. Identification Hypothesis Testing

Although the previous section has preliminarily confirmed that the implementation of green mergers and acquisitions by heavy polluters promotes corporate environmental performance, this result may still be disturbed by self-selection bias, omitted variables and other interferences that may make the results of the study unreliable, so we need to carry out the identification of the hypothesis test, including the parallel trend test, the counterfactual test, and the placebo test.

### 4.3.1. Parallel Trend Test

Referring to McGavock [57], a dynamic effects model is developed to consider the implementation of green M&As by heavily polluting firms as a shock event and to test for parallel trends in the dynamic effects of this shock. Specifically, it is only necessary to replace the interaction term of model (1) with several years before and after the implementation of the shock event, while other variables remain unchanged. In this paper, we construct a model of the dynamic effect of a green M&A for *j* years before and after the implementation of a green M&A; meanwhile, to avoid the problem of full covariance, we set 1 year before the implementation of a green M&A as the reference group, as shown in model (2):

$$EP_{i,t} = \sum_{j=1}^{5} \alpha_{-j} G_{-j} + \alpha_0 G_0 + \sum_{j=1}^{5} \alpha_j G_j + \lambda Controls_{i,t} + \gamma_t + \mu_i + \eta_k + v_f + \varepsilon_{i,t} \quad (2)$$

where *j* = 1, 2, 3, 4, 5. If *j* > 5, the sample data are classified as *j* = 5; if −*j* < −5, the sample data are classified as −*j* = −5. Since the listed heavy polluters did not implement green mergers and acquisitions at the same time, and not all sample firms carried out green mergers and acquisitions, the value of *G* is different in different firms.

Figure 1 reports the dynamic effect of the shock and its confidence intervals. The coefficient *α* fails the test of significance at the 5 percent level for the 2–5 years prior to the shock. This indicates that the changing trend of the treatment and control groups passed the parallel trend test. In other words, the significant difference between the environmental performance of the treatment and control groups after the implementation of green M&A by the enterprises is the result of the impact of green M&A rather than originating from ex ante differences. In addition, the coefficient *α* still has a significant downward trend 1 year after the M&A occurred, indicating that the impact of the implementation of green M&A by heavy-polluting enterprises on environmental performance has a certain degree of time persistence.

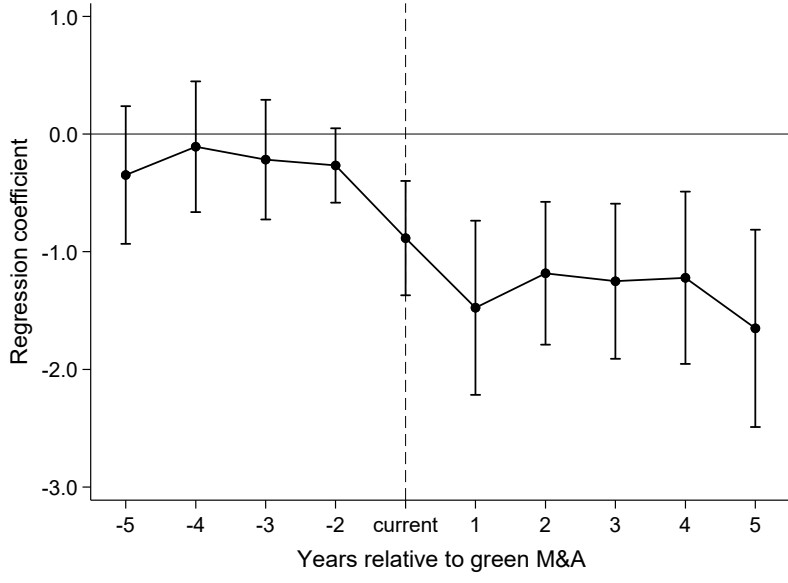

**Figure 1.** Parallel trend test. Note: Dashed lines perpendicular to the horizontal axis indicate 95 percent confidence intervals.

### 4.3.2. Counterfactual Tests

To exclude the impact of other firms' internal decision shocks or external policy disturbances on firms' environmental performance to ensure the stochasticity of the timing of green M&A implementation, this paper further conducts a counterfactual test on the

timing of green M&A implementation, as shown in Table 4. This is carried out by setting the time of implementation of green M&A to 1–2 years earlier and 1–2 years later, respectively, and then regressing sequentially. If the change in the time of M&A implementation can no longer continue to significantly affect the environmental performance of enterprises, it proves that the improvement of the environmental performance of heavy-polluting enterprises is caused by the implementation of green M&A decisions, and vice versa, it shows that the implementation of green M&A can not improve the environmental performance of enterprises.

**Table 4.** Counterfactual tests.

| Variables | (1) 2 Years in Advance | (2) 1 Years in Advance | (3) 1 Year Delay | (4) 2 Year Delay |
|---|---|---|---|---|
| *Greenma* | 0.300 | −0.141 | −1.057 *** | −1.120 |
| | (1.101) | (−0.573) | (−4.053) | (−1.489) |
| *Controls* | Yes | Yes | Yes | Yes |
| Individual FE | Yes | Yes | Yes | Yes |
| Time FE | Yes | Yes | Yes | Yes |
| _cons | 14.743 *** | 14.745 *** | 14.698 *** | 14.762 *** |
| | (90.292) | (90.969) | (95.666) | (94.206) |
| N | 1833 | 1833 | 1833 | 1833 |
| $R^2$ | 0.096 | 0.094 | 0.117 | 0.101 |

Note: Robust t-statistics in parentheses; *** $p < 0.01$, ** $p < 0.05$, * $p < 0.1$.

The results in Table 4 show that the coefficient of *Greenma* is insignificant when setting the time of green M&A implementation 1 year ahead, 2 years ahead, or 2 years behind, and the coefficient of *Greenma* is significantly negative only when setting the time of green M&A implementation pushed back by 1 year. This excludes the influence of other internal and external shock events of the firm and proves the robustness of the benchmark regression results in this paper. At the same time, since the sample of firms pushed back by 1 year is still significantly negative, this suggests that there is a certain lag in the effect of green M&As on firms' environmental performance in heavily polluting firms. This lag is not difficult to understand, because after the implementation of green M&As, enterprises need a certain amount of time to integrate the resources of the main M&A enterprise and the acquired enterprise to play the value of the M&A, and thus, for a while after the M&A occurs, the gradual internal integration of the enterprise and the operation is more smooth, and the effect of the improvement of the enterprise's environmental performance is more obvious.

### 4.3.3. Placebo Test

To rule out problems such as omitted variable interference and standard error bias arising from serial correlation, and to test the reliability of the baseline regression, we refer to Chetty et al. [58] for the placebo test. We obtain the distribution of regression results for the dummy samples by repeating the regression for 500 randomly sampled unduplicated samples through the nonparametric replacement test method, as shown in Figure 2. The distribution plot of the placebo test shows that the estimated coefficients of the random dummy samples are around the value of 0 and normally distributed, and the estimated coefficients of the benchmark regression (−0.968) fall within the small probability interval of the distribution of the coefficients of the nonparticipating permutation test, passing the placebo test, which reconfirms the robustness of the results of our study.

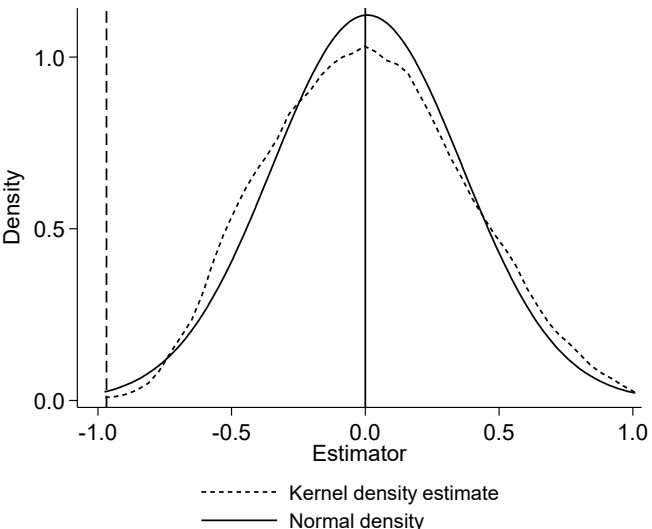

**Figure 2.** Empirical cumulative distribution of placebo test coefficients. Note: The solid line is the probability density distribution of the DID coefficients corresponding to the dummy samples, the dashed line is the normal distribution, the vertical solid line is a value of 0, and the vertical dashed line is the baseline regression estimate coefficient.

### 4.4. Robustness Analysis

Use of proxy variables for corporate environmental performance. In this paper, two methods are selected to standardize the treatment of corporate environmental capital expenditure as a proxy variable for corporate environmental performance: (i) As the financial position of the enterprise may have an impact on the amount of its environmental investment, the total corporate assets is one of the important static accounting elements reflecting the financial position of the enterprise, so this paper adopts the ratio of the corporate environmental capital expenditure to the total assets at the end of the year ($EP_1$) as a proxy variable for measuring environmental performance. (ii) Since the balance of shareholders' equity excludes total liabilities on the basis of total corporate assets, it may have a greater impact on environmental capital expenditure, so this paper further adopts the ratio of corporate environmental capital expenditure to the balance of shareholders' equity ($EP_2$) as a proxy variable for measuring environmental performance. The regression results are shown in columns (1) and (2) of Table 5, respectively, and the results are significantly negative, proving that the empirical results of this paper are robust.

**Table 5.** Robustness analysis.

| Variables | (1) $EP_1$ | (2) $EP_2$ | (3) M&A | (4) Capital City | (5) CEO Personal Traits | (6) | (7) Green Industrial Policies |
|---|---|---|---|---|---|---|---|
| *Greenma* | −0.102 * | −0.323 * | −1.069 *** | −1.095 *** | −0.971 *** | −0.957 *** | −0.786 ** |
| | (−1.783) | (−1.765) | (−3.839) | (−3.516) | (−3.761) | (−3.695) | (−2.311) |
| *Controls* | Yes | Yes | Yes | Yes | Yes | Yes | Yes |
| Individual FE | Yes | Yes | Yes | Yes | Yes | Yes | Yes |
| Time FE | Yes | Yes | Yes | Yes | Yes | Yes | Yes |
| *_cons* | 9.095 ** | 5.450 | 2.083 | 4.486 | 0.962 | 0.928 | −0.522 |
| | (2.556) | (1.462) | (0.604) | (0.873) | (0.337) | (0.330) | (−0.180) |
| N | 1833 | 1833 | 1194 | 507 | 1833 | 1833 | 1269 |
| $R^2$ | 0.336 | 0.041 | 0.194 | 0.174 | 0.186 | 0.194 | 0.193 |

Note: Robust t-statistics in parentheses; *** $p < 0.01$, ** $p < 0.05$, * $p < 0.1$.

Excluding the impact of non-merger and acquisition events. Considering that under the same year, firms with M&As may be significantly different from firms without M&As in terms of firm size, operating conditions, etc., this paper further excludes the non-M&A event samples and retains only the M&A samples to regress again, and the results are shown in column (3) of Table 5.

Consider the impact of regional factors. Considering that the level of economic development and the overall level of society in provincial capital cities is often higher than that in other cities, it may lead to differences between cities. On the one hand, the environmental regulation of local governments in provincial capital cities is higher than that in other cities, and the strategic choices of enterprises in provincial capital cities may differ significantly from those in other cities; on the other hand, the number of enterprises in provincial capital cities tends to be larger, and the base number of green M&A occurring is larger. Therefore, this paper retains only the samples of provincial capital cities and municipalities to regress the test again, and the results are shown in column (4) of Table 5.

Consider the impact of corporate CEO personal traits on corporate mergers and acquisitions. Psychological research suggests that managers' personal traits can greatly influence corporate strategic decisions. As one of the major business activities and capital operation methods of enterprises, M&As are directly affected by the core decision makers of enterprises. In this paper, to exclude the interference of the CEO's personal traits on corporate green M&As, the CEO's gender, age, education, salary, number of shareholdings, financial background, academic background, and overseas background are added as control variables on the basis of the original ones, in which column (5) of Table 5 adds only the first five CEO-level variables mentioned above, and column (6) adds all the eight trait variables mentioned above.

Removing the impact of green industrial policies on green M&A. To address the phenomenon of "greening" and spend limited funds and resources on the right things, China's National Development and Reform Commission (NDRC) and seven other ministries jointly issued the "Green Industry Guidance Catalogue (2019 Edition)" in March 2019, which clearly defines the boundaries of the green industry and points out the focus of development for the first time. The promulgation of the Green Industry Guidance Catalogue has an important guiding role and incentive effect on the green M&A decisions of heavily polluting enterprises. We exclude the data from 2019 to conduct the regression test again to exclude the impact of this external shock on corporate decision-making, and the results are shown in column (7) of Table 5.

*4.5. Elimination of Endogeneity Issues*

If the impact of green M&As of heavy polluters on firms' environmental performance is due to the omission of an unobserved variable that may affect both the green M&A decisions and environmental performance changes of heavy polluters, it will lead to a pseudo-correlation between green M&As and environmental performance, which will result in an endogeneity problem. To avoid the results of this paper from being affected by potential omitted variables and generating the endogeneity problem, this paper further addresses it by using PSM-DID and controlling for potential omitted variables.

In this paper, the PSM-DID method is used to rematch the sample using firm-level data on control variables to address the bias in the findings due to systematic differences in the variables. The results of another regression using the matched samples are shown in column (1) of Table 6, which is still significantly negative, proving the robustness of the benchmark regression. Based on the balance test results (as shown in Table 7), it can be seen that the standard deviations of the matched variables are all within 10% and the t-test does not reject the original hypothesis. Meanwhile, according to the graph of the common range of matching values (shown in Figure A1 in Appendix A) and the graph of the difference in variables before and after matching (shown in Figure A2 in Appendix A), the variables do not lose many samples after matching and the standard deviation is significantly reduced. This indicates that the balance test is passed and the endogeneity problem due to selection bias can be eliminated.

**Table 6.** Endogeneity test.

| Variables | (1) EP | (2) EP | (3) EP | (4) EP |
|---|---|---|---|---|
| *Greenma* | −1.028 *** | −1.062 *** | −0.938 *** | −1.039 *** |
| | (−5.266) | (−4.059) | (−2.932) | (−3.603) |
| *Controls* | Yes | Yes | Yes | Yes |
| Individual FE | Yes | Yes | Yes | Yes |
| Time FE | Yes | Yes | Yes | Yes |
| Province × year FE | No | Yes | No | Yes |
| Industry × year FE | No | No | Yes | Yes |
| _cons | 15.911 *** | −547.215 * | −185.949 | −515.055 |
| | (30.637) | (−1.923) | (−1.125) | (−1.475) |
| N | 1776 | 1833 | 1833 | 1833 |
| $R^2$ | 0.244 | 0.204 | 0.158 | 0.233 |

Note: Robust t-statistics in parentheses; *** $p < 0.01$, ** $p < 0.05$, * $p < 0.1$.

**Table 7.** Matching balance test results.

| Variable | Unmatched Matched | Mean Treated | Control | Bias (%) | \|Bias\| Reduct (%) | t | $p > \|t\|$ |
|---|---|---|---|---|---|---|---|
| *Size* | U | 42.474 | 47.594 | −45.8 | | −5.74 | 0.000 |
| | M | 42.650 | 43.460 | −7.2 | 84.2 | −0.59 | 0.558 |
| *Employees* | U | 11.895 | 7.289 | 34.7 | | 5.01 | 0.000 |
| | M | 11.693 | 11.109 | 4.4 | 87.3 | 0.31 | 0.756 |
| *Lev* | U | 0.562 | 0.522 | 18.2 | | 2.01 | 0.045 |
| | M | 0.561 | 0.562 | −0.4 | 98.1 | −0.03 | 0.977 |
| *Roe* | U | 7.911 | 6.806 | 29.0 | | 3.29 | 0.001 |
| | M | 7.854 | 7.661 | 5.1 | 82.5 | 0.40 | 0.690 |
| *Sale* | U | 52.667 | 48.503 | 71.4 | | 7.55 | 0.000 |
| | M | 52.644 | 52.241 | 6.9 | 90.3 | 0.63 | 0.527 |
| *CR* | U | 50.850 | 63.044 | −16.7 | | −1.82 | 0.069 |
| | M | 51.142 | 53.331 | −3.0 | 82.1 | −0.33 | 0.741 |
| *Lerner* | U | 3.240 | 3.530 | −32.9 | | −3.09 | 0.002 |
| | M | 3.242 | 3.240 | 0.2 | 99.4 | 0.02 | 0.986 |
| *Pgdp* | U | 0.007 | 0.038 | −21.3 | | −1.99 | 0.047 |
| | M | 0.007 | 0.011 | −3.2 | 85.2 | −0.42 | 0.673 |

Considering that other factors in both the region and the industry to which the firm belongs may affect the firm's M&A decision and environmental performance, we include province–year and industry–year interaction fixed effects in the baseline model, respectively, in order to control for the effects of time-varying potential omitted variables at the region and industry levels on heavily polluting firms. The regression results are shown in columns (2) to (4) of Table 6, and the results remain significantly negative. This suggests that the implementation of green M&As by heavily polluting firms does significantly reduce environmental expenditure capital and improve firms' environmental performance after accounting for potential influences at the regional and industry levels.

After re-validation with parallel trend tests, counterfactual tests, placebo tests, robustness tests, and endogeneity tests, we confirm that the results of the benchmark regression are robust. This result suggests that the implementation of green M&A by listed heavy polluting firms in China can significantly increase the environmental performance of the firms. Hypothesis 1 is confirmed and this conclusion is robust.

## 5. Heterogeneity Analysis

This paper examines the effect of corporate green M&As on environmental performance in terms of three dimensions: the nature of the firm, the age of the firm, and the fiscal environmental expenditure in the region where the firm is located.

The nature of the enterprise affects the effectiveness of green M&As. Compared with private enterprises that focus more on profit maximization, SOEs tend to take on more social responsibility and respond more actively to government policies related to green development [59]. Therefore, the role of environmental performance enhancement by SOEs implementing green M&As will also be greater. The results are shown in columns (1) and (2) of Table 8, with coefficients of −1.545 for SOEs and −0.552 for private firms. This suggests that green M&As by SOEs are more likely to achieve the purpose of enhancing environmental performance.

**Table 8.** Heterogeneity analysis.

| Variables | (1) Private Enterprises | (2) State-Owned Enterprises | (3) Mature Enterprises | (4) Young Enterprise | (5) High Environmental Investment | (6) Low Environmental Investment |
|---|---|---|---|---|---|---|
| *Greenma* | −0.552 ** | −1.545 ** | −0.932 * | −1.925 ** | −0.597 | −1.133 *** |
| | (−2.507) | (−2.585) | (−1.661) | (−2.183) | (−1.386) | (−2.928) |
| *Controls* | Yes | Yes | Yes | Yes | Yes | Yes |
| Individual FE | Yes | Yes | Yes | Yes | Yes | Yes |
| Time FE | Yes | Yes | Yes | Yes | Yes | Yes |
| *_cons* | 15.006 *** | 14.013 *** | 14.732 *** | 14.983 *** | 14.247 *** | 14.907 *** |
| | (75.987) | (49.057) | (94.652) | (49.140) | (61.815) | (69.081) |
| N | 1027 | 806 | 897 | 936 | 676 | 1157 |
| R$^2$ | 0.102 | 0.178 | 0.111 | 0.160 | 0.118 | 0.122 |
| *p*-value for coefficient difference | 0.003 | | 0.003 | | 0.028 | |

Note: *p*-values for coefficient differences were calculated from the estimates of the Chow test for the interaction term model. Robust t-statistics in parentheses; *** $p < 0.01$, ** $p < 0.05$, * $p < 0.1$.

For the test of heterogeneity of firms' age, see columns (3) and (4) of Table 8. We use the mean firm age to divide the study sample into two groups: mature and young firms. Compared to mature firms, young firms are mostly smaller and cash-strapped and need to choose their direction carefully and spend their limited time and resources where they are most needed. As a result, young companies are more motivated to implement M&As and expand their scale. They prefer to acquire ready-made technology and talent quickly and directly through M&As in order to realize the green transformation of their enterprises.

Local governments often use two complementary pathways to achieve environmental governance: fiscal inputs and regulation of enterprises. When financial resources are relatively scarce, local governments will find it difficult to maintain a high level of specialized expenditure on environmental protection, and can only exert greater pressure on enterprises to increase green investment and reduce pollution from production activities [60]. Therefore, in regions with smaller local financial investments in energy saving and environmental protection, the local government exerts more pressure on corporate environmental governance and is better able to guide and incentivize firms to improve corporate green performance through the implementation of green M&As. To examine the complementary effects of the above two governance pathways, this paper divides the study sample into two groups by using the median energy-saving and environmental protection expenditures in regional public finance expenditures each year. The results are shown in columns (5) and (6) of Table 8, and the results confirm that in the group of regions with lower financial

expenditures on energy conservation and environmental protection, the positive impact of corporate green M&As on environmental performance is greater.

## 6. Mechanism Analysis

### 6.1. Green Technology Innovation

Green mergers and acquisitions by heavily polluting enterprises can improve their environmental performance by promoting their green technological innovation. On the one hand, green M&As can significantly reduce the risk of enterprise green technological innovation, thus enhancing green technological innovation. Knowledge is an important source of competitive advantage for enterprises, but for a long time, China has overly relied on the crude development model, and the experience accumulated in the past is mainly concentrated in the field of high-pollution, high-energy-consumption or high-emission production, which makes it easy to form a traditional mindset lock within the enterprise, and it is difficult to break through the existing knowledge base to carry out green technological innovation [61,62]. The target companies of green M&As are carefully selected acquisition targets of heavy polluters with certain green sustainable development advantages. Through green M&As, the acquirer can effectively absorb the experience accumulated by the target company in the fields of green production, green service and green management, help itself overcome the various technical bottlenecks encountered in the process of green innovation, make up for the green technological deficiencies before the merger and acquisition, thus reducing the risk of green technological innovation of the enterprise, and thus rapidly improve the environmental performance of the enterprise after the merger and acquisition. On the other hand, green mergers and acquisitions can effectively improve the efficiency of enterprise green technology innovation, thus enhancing green technology innovation. If the green technology innovation is carried out independently, the heavy-polluting enterprises need to configure from scratch the various machines, equipment and professional and technical talents involved in the innovation process, from the procurement and installation of equipment to the training of personnel, which will be a time-consuming and labor-intensive upfront investment [63–65]. However, since the subject enterprise has already reserved the necessary green innovation facilities and talents in the field of green technology innovation, the main merger enterprise can acquire these resources in a short period through successful merger and acquisition activities, rapidly carry out green technology innovation activities, enhance the efficiency of the enterprise's green technology innovation, and realize the resource advantages of both parties in the transaction to complement each other so as to achieve the purpose of rapidly improving environmental performance.

To deeply explore the role of green technology innovation in the process of green mergers and acquisitions affecting environmental performance, this paper constructs the following model to test the impact mechanism of green mergers and acquisitions of heavy-polluting enterprises on the environmental performance of enterprises through enterprise green technology innovation.

$$Innovation_{i,t} = \alpha Greenma_{i,t} + \lambda Controls_{i,t} + \gamma_t + \mu_i + \varepsilon_{i,t} \tag{3}$$

$$EP_{i,t} = \alpha_1 Greenma_{i,t} + \alpha_2 Innovation_{i,t} + \lambda Controls_{i,t} + \gamma_t + \mu_i + \varepsilon_{i,t} \tag{4}$$

where *Innovation* is the green technological innovation of enterprises. Due to the stability and objectivity of the patent granting criteria and the availability of relevant data, the number of patents is therefore a very reliable indicator and better reflects the level of innovation. Referring to Ma et al. [45], this paper adopts the total number of patent applications in category Y02 of heavily polluted listed companies in the year as a proxy variable for their corporate green innovation. Specifically, the number of green invention patent applications is retained only for patents filed, granted and undisputed by Chinese citizens in China under the six categories of technologies related to climate change mitigation (Y02) category of the Cooperative Patent Classification (CPC). Model (3) represents the effect of corporate green M&A on corporate green technology innovation, and in model (4), $\alpha_1$ represents

the direct effect of corporate green M&A on corporate environmental performance, and $\alpha_1 \times \alpha_2$ represents the indirect effect of corporate green M&A on corporate environmental performance through green technology innovation. The results of the mechanism test are shown in Table 9.

**Table 9.** Mediating effect analysis.

| | (1) | (2) |
|---|---|---|
| **Variables** | *Innovation* | *EP* |
| *Greenma* | 0.601 * | −1.365 *** |
| | (1.874) | (−7.473) |
| *Innovation* | | −0.024 *** |
| | | (−2.886) |
| *Controls* | Yes | Yes |
| Individual FE | Yes | Yes |
| Time FE | Yes | Yes |
| *_cons* | −0.266 | 14.755 *** |
| | (−0.945) | (121.399) |
| N | 1833 | 1833 |
| $R^2$ | 0.091 | 0.243 |

Note: Robust t-statistics in parentheses; *** $p < 0.01$, ** $p < 0.05$, * $p < 0.1$.

In model (3), the regression coefficient of *Greenma* is significantly positive, indicating that the green M&A of heavy-polluting enterprises will significantly promote the green technology innovation of heavy-polluting enterprises. On the one hand, it is because through green M&A, heavy-polluting enterprises can quickly obtain green technology and equipment and other resources, improve energy saving and emission reduction ability, and enhance their environmental performance; on the other hand, through the introduction of green standards, heavy-polluting enterprises can not only directly improve the status quo of production and pollution, but also accelerate the emergence of a win–win situation of profitability and environmental protection with the catalytic effect of new technology and new talent, further accelerating the enhancement of environmental performance. In model (4), the regression coefficient of the intermediary variable innovation is significantly negative, indicating that the enhancement of enterprise green technology innovation can effectively reduce the enterprise's environmental protection investment expenditures and improve the enterprise's environmental performance. Therefore, the green mergers and acquisitions of heavy-polluting enterprises will effectively improve the environmental performance of enterprises by promoting the green technological innovation of enterprises. This finding confirms hypothesis 2.

*6.2. Corporate Internal Control Quality*

In order to verify the moderating effect of the quality of internal control in a firm, we constructed the following model:

$$EP_{i,t} = \alpha Greenma_{i,t} + \beta DIB_{i,t} + \theta(Greenma_{i,t} \times DIB_{i,t}) + \lambda Controls_{i,t} + \gamma_t + \mu_i + \eta_k + v_f + \varepsilon_{i,t} \tag{5}$$

where *DIB* is the quality of internal control and the data is obtained from the *DIB* internal control and risk management database. The validation results are shown in column (1) of Table 10, where the coefficient of $DIB \times Greenma$ is significantly negative at the 10% significance level. This indicates that the higher the quality of firms' internal control, the more the implementation of green M&As by heavily polluting firms enhances environmental performance, which confirms hypothesis 3.

**Table 10.** Regulatory mechanism test.

| Variables | (1) EP | (2) EP |
|---|---|---|
| *Greenma* | −0.822 *** | −1.035 *** |
| | (−4.179) | (−7.095) |
| *DIB* × *Greenma* | −0.001 * | |
| | (−1.871) | |
| *DIB* | 0.001 *** | |
| | (5.494) | |
| *CEOE* × *Greenma* | | −0.815 * |
| | | (−1.708) |
| *CEOE* | | −0.279 * |
| | | (−1.708) |
| *Controls* | Yes | Yes |
| Individual FE | Yes | Yes |
| Time FE | Yes | Yes |
| *_cons* | 15.210 *** | 15.149 *** |
| | (116.501) | (167.342) |
| N | 1833 | 1833 |
| $R^2$ | 0.083 | 0.231 |

Note: Robust t-statistics in parentheses; *** $p < 0.01$, ** $p < 0.05$, * $p < 0.1$.

### 6.3. CEO Environmental Experience

We developed the following model to verify the moderating role of the CEO's environmental experience in the process of corporate green M&As affecting environmental performance:

$$EP_{i,t} = \alpha Greenma_{i,t} + \beta CEOE_{i,t} + \theta(Greenma_{i,t} \times CEOE_{i,t}) + \lambda Controls_{i,t} + \gamma_t + \mu_i + \eta_k + \upsilon_f + \varepsilon_{i,t} \tag{6}$$

where *CEOE* is the CEO's environmental experience. This paper manually collects and arranges the resumes of CEOs of each listed enterprise, and judges whether they have environmental protection experience according to whether they have worked in companies or departments in the environmental protection field in their early work experience, and whether they have obtained environmental protection related degrees, titles or patents. If it is judged that the CEO of the enterprise has environmental protection experience, it is assigned a value of 1; otherwise, it is 0.

The results are shown in column (2) of Table 10, where the coefficient of *CEOE* × *Greenma* is significantly negative at the 10% significance level. This suggests that the implementation of green M&As by heavily polluting firms is more likely to enhance the firm's environmental performance if the CEO of the firm has had environmental experience. The results of the above studies confirm hypothesis 4.

## 7. Conclusions and Policy Recommendations

For a long time, China has caused irreversible damage to the environment in order to promote rapid economic development. Promoting the green transformation of heavily polluting enterprises is the focus and difficulty of China to explore the mutual benefit road between economy and environment, to promote sustainable development, and to establish an environmental and resource-friendly society. Based on the panel data of Chinese listed companies in the heavy pollution industry from 2010 to 2022, this paper investigates the impact of the implementation of green M&As by heavy-polluting enterprises on their environmental performance by using the DID method. Our findings are as follows: (1) The implementation of green M&As by the listed Chinese heavy polluters can reduce corporate environmental capital expenditure and significantly improve corporate environmental performance; this result remains robust after a series of tests. Meanwhile, it is found that state-owned enterprises and young enterprises are more inclined to achieve technological transformation, emission reduction and green development through green M&As. And in regions with lower financial

expenditures on energy conservation and environmental protection, the positive impact of corporate green M&As on environmental performance is greater. (2) Through the mediating mechanism test, it is found that the green M&A of heavy-polluting enterprises will effectively enhance their environmental performance by promoting their green technological innovation. (3) After the test of the moderating effect, we found that when the quality of the internal control of the enterprise is higher, the implementation of green M&As by heavily polluting enterprises can better enhance the environmental performance of the enterprise. (4) Through further research, and when the CEO of the enterprise has had the experience of environmental protection, the implementation of green M&As by heavily polluting enterprises can better enhance the environmental performance of the enterprise.

Based on the above research findings, this paper puts forward the following policy recommendations:

The government should improve the environmental regulation policy system and promote green mergers and acquisitions by heavily polluting enterprises. Our findings suggest that green mergers and acquisitions can improve corporate environmental performance. The central and local governments should deeply implement the responsibility system for controlling environmental objectives and the environmental regulatory policy system, set strict limits on the total amount of pollutants and emission standards for heavily polluting enterprises, improve the cost of pollution violations by enterprises, force emission-control enterprises to implement green mergers and acquisitions, increase green investment, drive endogenous development through scientific and technological innovation, and be the driving force of intellectual capital to achieve the improvement of corporate environmental performance and green transformation.

Enterprises should actively alleviate environmental protection pressure through green M&As and focus on improving their green innovation ability and enterprise value. Our study confirms the mediating role of green technology innovation in the process of green M&As affecting firms' environmental performance. Heavily polluting enterprises should pay attention to green mergers and acquisitions in the process of green transformation, and learn from, study, and refer to the replicable and popular merger and acquisition cases and experiences of successful transformation enterprises. Especially for non-high-tech enterprises with shorter establishment time and smaller scale, green M&As should be taken as an important part of introducing cutting-edge technology, transforming development modes and creating opportunities for change. At the same time, enterprises that have implemented green M&As or are in the process of a green M&A should pay attention to the fact that the effect of the M&A is not long-lasting, and they need to take advantage of the resources and transformation opportunities brought by the M&A as soon as possible to continue in-depth development of green technological innovation, promote investment in cleaner production and truly realize substantial transformation and sustainable green development of enterprises.

The government should take measures to help executives of heavily polluting firms enhance their environmental awareness and curb the short-sighted behavior of corporate management by improving the firm's internal control mechanism. We find that firms with higher quality internal controls and firms with environmentally experienced CEOs enhance the positive impact of green M&As in terms of environmental performance. Therefore, for heavily polluting enterprises to achieve true green transformation, it is critically dependent on the awakening and enhancement of corporate executives' environmental awareness. At this stage, despite the fact that all sectors of society attach great importance to environmental pollution, most heavily polluting enterprises are still in a game with the public and government departments. In addition to a sound external monitoring mechanism, it is more important to think about how to reverse the inherent thinking of heavily polluting enterprises and enhance their awareness of proactive environmental protection and transformation and upgrading so that they can implement substantial green investment and industrial transformation. At the same time, improving the internal control mechanism is also conducive to reducing the short-sighted behavior of heavy polluters. A sound internal control mechanism can reduce the adverse impact on the long-term strategic

decision-making of enterprises due to the short-sightedness of the chairperson or general manager, which is conducive to improving scientific and sustainable decision-making and promoting the real green transformation of heavily polluting enterprises.

**Author Contributions:** Conceptualization, Y.X. and W.W.; methodology, Y.X.; software, Y.X.; validation, Y.X., W.W. and H.G.; formal analysis, H.Z.; investigation, H.G.; resources, Y.X.; data curation, W.W.; writing—original draft preparation, Y.X.; writing—review and editing, W.W.; visualization, H.G.; supervision, H.Z.; project administration, Y.X.; funding acquisition, H.G. All authors have read and agreed to the published version of the manuscript.

**Funding:** This research was supported by Hubei Provincial Department of Education: Major Projects of Philosophy and Social Science Research in Hubei Province (Grant No.: 21ZD010); Ministry of Education of the People's Republic of China: Central Universities Education Teaching Reform Project (Grant No.: BJ20220213); National Office for Philosophy and Social Sciences: Major Projects of the National Social Science Foundation of China (Grant No.: 23&ZD179); Zhongnan University of Economics and Law: Zhongnan University of Economics and Law Postgraduate Research and Innovation Programme (Grant No.: 202410305).

**Institutional Review Board Statement:** Not applicable.

**Informed Consent Statement:** All individuals taking part in the study gave their informed consent.

**Data Availability Statement:** The data and models used during the study are available from the corresponding author by request.

**Conflicts of Interest:** The authors declare no conflicts of interest.

## Appendix A

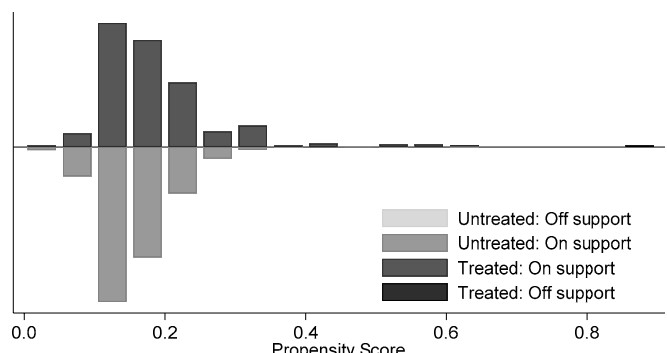

**Figure A1.** Common range of values for propensity score matching.

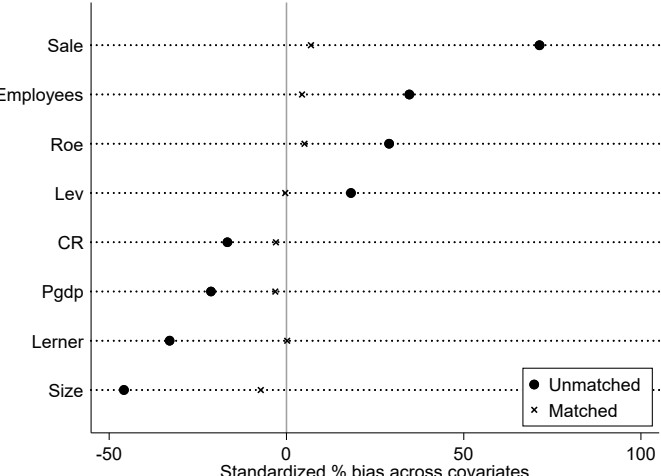

**Figure A2.** Difference in variables before and after PSM-DID matching.

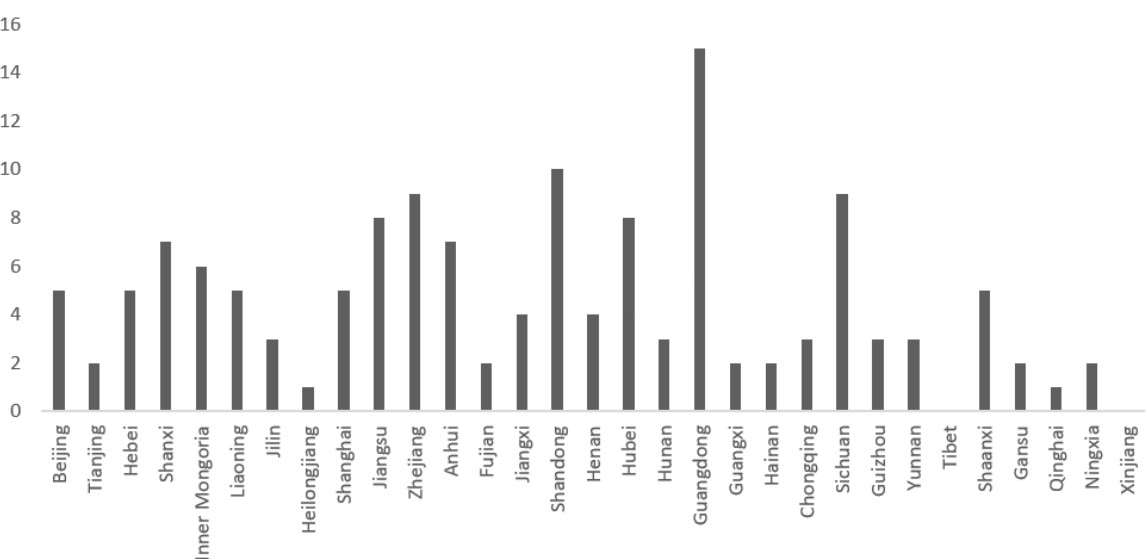

**Figure A3.** Sample Statistics of Enterprises. Note: Horizontal coordinates are provinces in China.

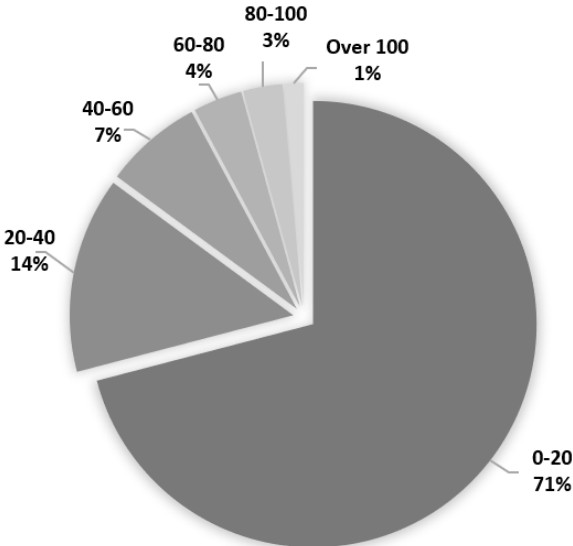

**Figure A4.** Statistical of the distribution of enterprise size zones as a percentage (total assets in billions).

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
