# Peer review of "The Impact of Green Mergers and Acquisitions on Corporate Environmental Performance: Evidence from China’s Heavy-Polluting Industries"

_sustainability, doi:10.3390/su16093796_

Round 1
Reviewer 1 Report
Comments and Suggestions for Authors
Comments for Authors
Title: The Impact of Green Mergers and Acquisitions on Corporate Environmental Performance: Evidence from China's Heavy Polluting Industries
Manuscript Number: sustainability-2954151
Article Type: Research Paper
General comments: In the manuscript Ying-ying et al. examined the impact of green mergers and acquisitions (green M&A) on corporate environmental performance. The results showed that the adoption of green M&A by Chinese listed heavy polluters can lower corporate environmental capital expenditure and significantly improve corporate environmental performance. In my opinion straight away the findings are interesting, the authors put contributions that the readers can believe to be worth reading. Overall, my recommendation is to be consider for publication in the said Journal.
Specific comments
1. Abstract: this part of the manuscript provided solid justification or suggestion on the basis of the authors thinking why this study was proposed. I would suggest the authors should just put some prominent findings.
3 Introduction and review literature: I read every single line which is presented well and organized, just my recommendation is to add recent literature because the references provided are not recent.
4 Novelty statement and Hypothesis development: the work has worth and can provide future direction for in-depth investigation.
5 Research design and Empirical results: the design of the work is excellent I don’t have any comments regarding the design of the study as well the results are justified and well presented
6 Overall recommendation. In my point of view there is no technical or critical error or deficiency in the work conducted. I am in the favor of recommendation of this paper.
Comments on the Quality of English LanguageCan be thouroughly checked from native acadamic speaker.
Author Response
We have read your comments carefully and thank you very much for your recognition of our paper. Two of your comments clearly illustrate the shortcomings of the article, and we have made changes to address these two issues.
- In response to your suggestion of put some prominent findings in the abstract section, we have made appropriate deletions and adjustments to the abstract section. It's in lines 8-25 of the manuscript.
- In response to your suggestion about citing the latest literature, we have added relevant literature from 2024 and highlighted it in the reference section.
Reviewer 2 Report
Comments and Suggestions for Authors
The manuscript constitutes an interesting empirical study and may be of interest to other researchers. A strength is the empirical study itself, which is well presented. Research hypotheses were correctly defined. The inference is not controversial, but it could be more developed. It should be indicated what the study contributes to the literature on the subject and what its importance is for society. The manuscript's shortcoming is the poor literature discussion. It should be strengthened with publications from Europe and North America (other authors outside China's area of influence). The authors will present the review, omitting significant research by other authors. These studies should be referred to in the literature discussion. Thus, the study will maintain balance and the study will have a good theoretical foundation.
The article needs to be improved.
Author Response
Thank you very much for your valuable suggestions on our article, which we have revised to address the two issues you mentioned.
- You suggested that the article should state the contribution of the study to the relevant literature and its importance to society. We have added the significance of the study to the existing literature and to the society in the Introduction section: “Firstly, existing literature tends to focus on the motivations of firms to green M&A and ignores the impact of green M&A. We examine the objective performance of green M&As affecting environmental performance of heavily polluting firms, expand the research on the influencing factors of corporate green M&As and green investments, enrich the research on the driving mechanisms of corporate environmental performance, and provide micro evidence for promoting the green development of heavily polluting firms … Thirdly, we provide useful advice for companies to formulate their growth strategies, meanwhile, the conclusions provide a theoretical basis and policy rationale for how the relevant government departments can further improve the governance mechanism for the corporate environment and promote the harmonious development of the environment and the economy, and also provide an effective reference for other developing countries.” It's in lines 59-74 of the manuscript.
- You suggested that the article should be strengthened by publications of other authors outside the Chinese sphere of influence. We have added relevant studies by authors outside of China's sphere of influence in the Literature review section to maintain the balance of the study’s foundatiaon, and highlighted the new literature one by one in the reference section.
Reviewer 3 Report
Comments and Suggestions for Authors
please define what you mean by narrow environmental performance, green companies, green merger, what is a "polluting industry"
Comments on the Quality of English Languagenone
Author Response
Thank you very much for your valuable suggestions on our article, we have added definitions of narrow environmental performance, green mergers and acquisitions, and polluting industries. We did not add a definition of "green company" as it was not mentioned in the article. The following are the relevant additions:
Narrow environmental performance: “while narrow environmental performance refers to a system of indicators that can be identified and measured by a company through quantitative criteria, for example, the quantitative levels of solid, liquid, gaseous and other types of harmful substances emitted by enterprises in the course of production and operation.” It's in lines 83-87 of the manuscript.
Green mergers and acquisitions (green M&A): “Green M&A refers to the M&A implemented by enterprises adhering to the green concept for the purpose of acquiring resources such as green technology and equipment, improving energy-saving and emission reduction capabilities, and realizing green transformation.” It's in lines 316-319 of the manuscript.
Polluting industries: “Meanwhile, according to the Circular of the Ministry of Ecology and Environment of the People's Republic of China on the Issuance of the Classification and Management List of Listed Companies in Environmental Protection Verification Industry, we define the following industries as heavy polluting industries: coal mining and washing industry, oil and gas mining, ferrous metal mining and processing industry, non-ferrous metal mining and processing industry, textile industry, leather, fur, feather and its products and footwear industry, paper and paper products industry, petroleum Processing, Coking and Nuclear Fuel Processing Industry, Chemical Materials and Chemical Products Manufacturing Industry, Chemical Fibre Manufacturing Industry, Rubber and Plastic Products Industry, Non-Metallic Mineral Products Industry, Ferrous Metals Smelting and Rolling Processing Industry, Non-Ferrous Metals Smelting and Rolling Processing Industry, and Electricity and Thermal Power Production and Supply Industry. According to the Guidelines on Industry Classification of Listed Companies revised by the China Securities Regulatory Commission in 2012, the codes for the heavy pollution industries are B06, B07, B08, B09, C17, C19, C22, C25, C26, C28, C29, C30, C31, C32 and D44.” It's in lines 278-292 of the manuscript.
Reviewer 4 Report
Comments and Suggestions for Authors
The paper presents a very intriguing idea since the author are correct that most of the research and the common understanding is that heavy polluting industries only have M&As with green industries to promote their image or ward off regulators. Heavy polluting industries are not the only ones to engage in this type of behavior. Companies with image problems will do something similar to improve corporate image. Such as when Philip Morris acquired Kraft only to promote the good work Kraft was already doing as their own.
· The article is very well written, the only things are saw were more stylistic difference not issues.
· The literature review does seem to be pretty through but does rely a little heavily on studies by Pan Et.al. (This may be due to lack of research in this area however).
· The sample selection and data section (3.1) could be improved. The authors never provide a number for how many companies were included in the sample, nor any demographic information about the companies included in the sample. Since so many of the variables include demographic data about the companies it would have been nice to information about items such as location in China (since GDP of the region is one of the variables), size of company (one of the variables) etc.
· And when they listed the means they used to screen the sample, they listed one method they used to exclude, two methods they used to include companies in the sample and then two more methods they used to exclude companies from the sample. It would be much cleaner if they listed all the inclusion criteria and all the exclusion criteria together. Also the two reasons for inclusion were confusing and could use clarification.
· The authors define the dependent variable (EP) as ‘the natural log of the firms’ environmental capital expenditure’ but the independent variable was a dummy of if the M&A is green or not and the rest of the variables listed were defined as control variables in the test. Not sure how a dummy variable can be the natural log of the firms’ environmental capital expenditure, this needs significant clarification.
· While the literature review also helped with good Hypothesis development, the authors seems to have forgotten to actually test the four hypotheses they proposed. After section 2.2 where they develop their hypotheses, the only other time they mention the word hypothesis is in section 4.3 where they carry out an identification of the hypothesis text. When testing hypotheses one needs to say more than ‘although the previous section has preliminarily confirmed that the importance of green mergers and acquisitions by heavy polluters promotes corporate environmental performance.’ (p.9) even the results they list in section 7 do not really connect to the hypotheses.
· The authors did a good job developing models and running said models with lots of different data points, but they seems to get caught up in this process and not on the clear results. If one runs enough tests on data, one can get it to say anything they wants. There are ways to account for missing variables and error bias without running a Placebo test.
· The authors could have come to their policy recommendations (which are good) without running so many tests on the data.
Author Response
Thank you very much for your comments, and we have revised the relevant parts of the article according to your suggestions.
- 1. You suggested that the article is more dependent on the study of Pan et al, and the existing literature really lacks the study in this area, but we still added some relevant literature as far as possible, and highlighted the newly added literature one by one in the reference section.
- You suggested that the article did not provide the number of sample companies and related statistical information, we added the number of sample companies (141) and the location of the sample companies in China, and the statistical results are shown in Figure 3 in the Appendix.
- You suggested that the article's presentation of the method of screening samples was unclear, and that both the retention and exclusion criteria should be listed together. We have reorganised the presentation of the data selection section. The content is as follows: “We screened the sample by: (1) Excluding M&A events with failed deals, acquisition amounts less than 1 million RMB, equity acquisition ratios less than 30% or already holding more than 30% equity ratio in the target company. (2) Excluding M&A samples with failed deals or missing data. (3) Excluding ST, PT and insolvent companies. (4) Excluding M&A samples where the business type is divestiture, asset replacement, debt restructuring, or share buyback. Finally, we retained only M&A events where the transaction type was an equity acquisition. At the same time, if the same firm conducted multiple M&As in the same year, the samples with the same M&A targets are combined, and only the sample with the largest transaction amount and the highest acquisition ratio is retained for the samples with different M&A targets. After screening, we obtained a total sample of 141 firms. Figure 3 in the Appendix reports the statistical description of the sample of firms.” It's in lines 264-277 of the manuscript.
- Your question " Not sure how a dummy variable can be the natural log of the firms’ environmental capital expenditure", we have carefully checked the article and found that the y variable is Environmental Performance (EP), which is characterised by the natural logarithm of an enterprise's environmental capital expenditures, and is not a dummy variable. The x variable is Green M&A (Greenma), which is a dummy variable.
- You pointed out that the manuscript neglects to test the four hypotheses we proposed. In response to your suggestion, we have adjusted the order of the empirical part and stated the hypotheses verified by the research in the corresponding part. Our experimental design corresponds to the theoretical hypotheses, and our conclusions should also correspond to the empirical section. According to your suggestion, the conclusions in part 7 are also adjusted to 4, which correspond to the 4 hypotheses we proposed and the 4 results verified by the empirical evidence. It's in lines 387, 558-562, 670, 682, 700, 710-726 of the manuscript.
Round 2
Reviewer 2 Report
Comments and Suggestions for Authors
Thank you for the improvement. I don't make any comments.
Author Response
Thank you for your affirmation.
Reviewer 4 Report
Comments and Suggestions for Authors
The authors did address most of my concerns to my satisfaction. However, the only demographics they provide are the number of companies in each region and nothing else. I also think they should double check the citation methods in the text and make sure they are consistant.
Author Response
Thank you very much for your professional suggestions, we have made revisions: 1. Added Figure 4 in the appendix for statistics on the size of the sample firms. 2. Checked the citation format in the article and revised the inconsistent citation methods.